# LabInsect-48K: A Comprehensive Dataset for Visual Insect Understanding

## Abstract

Visual understanding of insects from historical collections is crucial for insect bio-diversity, ecological sustainability, and agricultural management. However, most existing datasets mainly focus on semantic labels and lack the spatial annotations which are essential for real-world applications and morphological analysis. To address these limitations, we introduce **LabInsect-48K**, the first comprehensive dataset of high-resolution entomological specimen images sourced from museum archives. LabInsect-48K contains 48,400 images spanning 643 species across 4 major insect orders. Delivers comprehensive and precise annotations of insects in both semantic and spatial dimensions, with the aim of advancing the landscape of biodiversity research communities. Specifically, the dataset provides hierarchical taxonomic labels in semantics and supports different levels of categorization. More importantly, the dataset also has fine-grained spatial annotations to support quantitative analysis of morphology, *e.g.*, insect shape, size, and structural traits. Our dataset supports three core tasks for comprehensive understanding: species-level classification, object detection, and instance segmentation. We further demonstrate the dataset's potential for open-set discovery and cross-dataset generalization. LabInsect-48K thus serves as a cross-disciplinary resource: it facilitates accurate and fine-grained insect recognition and analysis for entomological research.

## 1 Introduction

Biodiversity plays a fundamental role in ecosystem resilience and long-term viability of natural and agricultural systems (Cardinale et al., 2012; Sala et al., 2000). Insects, comprising more than half of all known species (Stork, 2018), are central to ecological processes such as pollination, nutrient cycling, and food web stability (Losey & Vaughan, 2006). Their abundance and diversity make them key indicators for environmental change. Despite their ecological importance, the systematic understanding and monitoring of insect biodiversity remains a challenge, particularly on the global scale. The digitization of museum collections and the rapid advancement of computer vision technologies now offer an unprecedented opportunity to scale up insect identification, classification, and trait analysis through automation.

Recent efforts have introduced large-scale insect datasets, such as IP102 (Wu et al., 2019a) and BIOSCAN-5M (Gharaee et al., 2024) to accelerate machine learning research in biodiversity. While these datasets have facilitated progress in species-level classification, they primarily focus on semantic annotations and often lack spatial precision, *e.g.*, bounding boxes and instance masks. Some examples are shown in Figure 1 (a) and Figure 1 (b). These limitations restrict their utility in real-world applications where localization and morphological precision are necessary. For example, in trait-based analysis, automated taxonomy, and biodiversity informatics, variations in body size, specimen pose, and structural ratios are always relevant and helpful.

To address these challenges, we introduce LabInsect-48K, the first comprehensive, high-resolution benchmark dataset designed to support visual insect understanding based on morphology. LabInsect-48K contains 48,400 digitized specimen images spanning 643 species across four major insect orders. Each image in our dataset presents a single standardized specimen, photographed under controlled lighting and scale conditions. Each image is captured at high resolution, often exceeding $4000 \times 5000$ pixels. This setup ensures the detailed preservation of anatomical features such as antennae, legs, and wing venation. Some examples of our dataset are shown in Figure 1 (c).

Figure 1: Comparison between samples from previous datasets and our LabInsect-48K dataset. (a): IP102 (Wu et al., 2019a),(b) BIOSCAN-5M (Gharaee et al., 2024), (c) Our LabInsect-48K.

The annotations provided in our dataset span two dimensions: semantic and spatial. In the semantic dimension, we assign each specimen with hierarchical taxonomy, *i.e.*, order, genus and species. All labels are manually verified by domain experts to ensure taxonomic accuracy. These labels support flat and hierarchical classification tasks and enable genus-level generalization for rare or unseen species. In the spatial dimension, we first annotate each specimen with both bounding boxes to support spatial reasoning. Bounding boxes serve to localize insect regions and provide coarse structural context, enabling object detection models to distinguish between species based on position, size, and bounding-box-level aspect ratio. In ecological applications, instance-level detection allows scalable assessment of species richness, abundance, and distribution across space, making it a key component in automated biodiversity monitoring.

For finer-grained analysis, we provide pixel-level instance masks that capture precise specimen contours. These masks are essential for tasks such as segmentation, computational trait extraction, and morphological quantification. The segmentation goes beyond coarse object detection and allows more precise evaluation of model performance on detailed contour prediction. Traits like body length, wing area, mask-level aspect ratio, and compactness can be derived directly from the masks, supporting comparative taxonomy and phenotype-based clustering.

With the rich semantic and spatial labels on diverse insect corpus, LabInsect-48K jointly support fine-grained classification, spatial detection, and morphological segmentation of insects at scale. We benchmark representative models including ResNet-50 (He et al., 2016), HRNet-W32 (Sun et al., 2019), ViT-B/16 (Dosovitskiy et al., 2021), MobileNetV4-L (Qin et al., 2024), and zero-shot CLIP (Radford et al., 2021). For detection, we evaluate a broad spectrum of models, from classical detectors like Faster R-CNN (Ren et al., 2015) to advanced transformer-based approaches such as Deformable DETR (Zhu et al., 2020), DINO (Zhang et al., 2022). For instance segmentation, we consider both proposal-based and query-based methods, including Mask R-CNN (He et al., 2017), SOLOv2 (Wang et al., 2020c), and QueryInst (Fang et al., 2021). Moreover, we also perform experiments to study dataset's potential for open-set discovery and cross-dataset generalization. We emphasize that our primary objective is not to create a benchmark that is "hard" in the computer vision sense (e.g., occlusion, background clutter, domain shift), but rather to facilitate accurate, fine-grained, and interpretable insect recognition and analysis for entomological research. The main contributions are summarized as follows:

- **Dataset Construction:** We construct a new image-based insect dataset with rich semantic and spatial annotations for multiple tasks. To the best of our knowledge, LabInsect-48K is the first dataset with comprehensive annotations for visual insect understanding.

- **Three-level annotations:** Each image is labeled with order/genus/species taxonomy, bounding boxes, and pixel-level masks to enable detection and morphological analysis.

Table 1: Comparison of insect-related datasets. LabInsect-48K is distinguished by its expert-verified hierarchical taxonomy, and it is the only high-resolution insect dataset providing both bounding box and instance mask annotations at scale. Unlike most existing datasets that focus on classification, LabInsect-48K enables fine-grained spatial analysis, supporting detection and segmentation tasks critical for morphology-aware modeling.

| Dataset | Year | Species | Hierarchical Labels | Bounding Boxes | Masks | Number of Samples |
|---|---|---|---|---|---|---|
| Samanta et al. (Samanta & Ghosh, 2012) | 2012 | 8 | ✗ | ✗ | ✗ | 609 |
| Wang et al. (Wang et al., 2012) | 2012 | 221 | ✓ | ✗ | ✗ | 225 |
| Venugoban et al. (Venugoban & Ramanan, 2014) | 2014 | 20 | ✗ | ✗ | ✗ | 200 |
| Xie et al. (Xie et al., 2015) | 2015 | 24 | ✗ | ✗ | ✗ | 1,440 |
| Liu et al. (Liu et al., 2016) | 2016 | 12 | ✗ | ✓ | ✗ | 5,136 |
| Xie et al. (Xie et al., 2018) | 2018 | 40 | ✗ | ✗ | ✗ | 4,500 |
| Deng et al. (Deng et al., 2018) | 2018 | 10 | ✗ | ✓ | ✗ | 563 |
| Alfarisy et al. (Alfarisy et al., 2018) | 2018 | 13 | ✗ | ✗ | ✗ | 4,511 |
| PestNet (Liu et al., 2019) | 2019 | 40 | ✗ | ✓ | ✗ | 88,670 |
| IP102 (Wu et al., 2019a) | 2019 | 102 | ✓ | ✗ | ✗ | 75,222 |
| Pest24 (Wang et al., 2020a) | 2020 | 24 | ✗ | ✓ | ✗ | 25,000 |
| AgriPest (Wang et al., 2021) | 2021 | 14 | ✓ | ✓ | ✗ | 49,707 |
| INSECT (Badirli et al., 2021) | 2021 | 1,213 | ✓ | ✗ | ✗ | 21,212 |
| iNat-2021 (Van Horn et al., 2021) | 2021 | 2,752 | ✓ | ✗ | ✗ | 723,816 |
| Insect-1M (Nguyen et al., 2024) | 2023 | 34,212 | ✓ | ✗ | ✗ | 1,017,036 |
| BIOSCAN-1M (Gharaee et al., 2023) | 2023 | 90,918 | ✓ | ✗ | ✗ | 1,128,000 |
| BIOSCAN-5M (Gharaee et al., 2024) | 2024 | 324,411 | ✓ | ✗ | ✗ | 5,150,000 |
| **LabInsect-48K (Ours)** | 2025 | 643 | ✓ | ✓ | ✓ | 48,400 |

- **Benchmarking:** We benchmark representative models across classification, detection, and segmentation. Meanwhile, we study dataset's potential for open-set discovery and cross-dataset generalization to show its value to entomological research.

## 2 RELATED WORK

### 2.1 INSECT DATASETS AND TAXONOMIC CLASSIFICATION

Insect datasets have played an essential role in applying machine learning to entomology and biodiversity research. Early datasets, such as those by Samanta et al. (Samanta & Ghosh, 2012), Wang et al. (Wang et al., 2012), and Xie et al. (Xie et al., 2015), were limited in scale, offering fewer than 50 species and lacking spatial annotations. IP102 (Wu et al., 2019a) introduced a more diverse benchmark with 102 pest categories, but it includes only image-level labels under variable field conditions. BIOSCAN-5M (Gharaee et al., 2024) represents a recent large-scale effort, offering millions of images and DNA barcodes, but lacks spatial labels such as bounding boxes or masks.

As summarized in Table 1, most existing datasets focus more on species classification and only a few of them consider the support of detection. Our proposed LabInsect-48K addresses these limitations by providing expert-verified hierarchical taxonomy (order, genus, and species), and adding bounding boxes and pixel-level instance masks for each specimen. With 643 species and over 48,000 curated images from museum archives, LabInsect-48K supports comprehensive visual understanding tasks beyond classification and lay the foundation for trait analysis and spatial reasoning.

To evaluate classification performance, we benchmark five representative models spanning convolutional and transformer-based backbones. ResNet-50 (He et al., 2016) and HRNet-W32 (Sun et al., 2019) serve as strong CNN baselines with varied feature fusion strategies. ViT-B/16 (Dosovitskiy et al., 2021) offers a global attention-based architecture for patch-wise encoding. MobileNetV4-L (Qin et al., 2024), a lightweight yet powerful convolutional model, supports deployment in edge scenarios. We also assess zero-shot classification using CLIP (Radford et al., 2021), which aligns image and text embeddings in a shared space. The results on these models offer insights into generalization across taxonomic levels and morphological variation.

### 2.2 OBJECT DETECTION METHODS

Object detection is a fundamental task in computer vision, aiming to localize and classify individual objects within an image (Zou et al., 2023; Khan et al., 2022; Xu et al., 2022; Zhang et al., 2023; Zhou et al., 2022; Xiong et al., 2021). Conventional detectors can be divided into two main categories:

two-stage and one-stage methods. Two-stage detectors like Faster R-CNN (Ren et al., 2015) remain a strong baseline, leveraging region proposals and ROI-based refinement. Mask R-CNN (He et al., 2017), while primarily designed for segmentation, also yields competitive detection performance through its shared backbone and classification head. One-stage detectors, like RetinaNet (Lin et al., 2017) and YOLO (Redmon et al., 2016), unify the detection pipeline into a single forward pass, offering higher speed but typically lower accuracy on small objects. YOLOX (Ge et al., 2021) provide efficient detection with decoupled heads and anchor-free design, which is particularly effective for dense or small targets.

Transformer-based architectures have significantly reshaped object detection paradigms. Deformable DETR (Zhu et al., 2020) enhances the convergence and scale adaptability of DETR (Carion et al., 2020) through deformable attention modules. DINO (Zhang et al., 2022) improves training stability and query usage for dense prediction. ViTDet (Li et al., 2022) adopts vision transformers to build a fully convolution-free detector with strong global reasoning capability. More recently, DiffusionDet (Chen et al., 2023) incorporates denoising diffusion models into the detection process, generating object proposals via iterative refinement.

Beyond closed-set settings, open-vocabulary detection has emerged to handle unseen categories. Models like ViLD (Gu et al., 2021), GLIP (Li* et al., 2022), and Grounding DINO (Liu et al., 2023) align visual features with large-scale language models or CLIP-style embeddings (Radford et al., 2021), enabling zero-shot detection via natural language prompts. These models are particularly relevant in biodiversity domains where taxonomic categories are continuously expanding.

In our benchmark, we include both CNN-based and transformer-based closed-set methods and open-vocabulary models to evaluate their performance in detecting insects of varying scale, morphology, and taxonomic diversity. The diversity and resolution of LabInsect-48K make it a robust platform for advancing detection models.

### 2.3 INSTANCE SEGMENTATION METHODS

Instance segmentation aims to predict individual object masks at the pixel level, enabling spatially detailed understanding of object morphology (Hafiz & Bhat, 2020; Chen et al., 2020; Cheng et al., 2022b; Vu et al., 2021). Classical approaches like Mask R-CNN (He et al., 2017) extend region-based detectors with a segmentation branch, producing high-quality instance masks. Recent research has proposed instance-aware segmentation methods that decouple mask prediction from box proposals. SOLOv2 (Wang et al., 2020c) formulates segmentation as location-based mask classification, while Decoupled-SOLO (Wang et al., 2020b) enhances this with refined feature decoupling for better contour description. CondInst (Tian et al., 2020) and BoxInst (Tian et al., 2021) further streamline the instance segmentation pipeline by conditioning dynamic mask heads on shared features or box-level supervision, reducing computational overhead and improving annotation efficiency.

Transformer-based segmentation frameworks like QueryInst (Fang et al., 2021) integrate instance-aware queries into deformable transformer backbones to unify detection and segmentation in a query-centric design. Additionally, ViTDet (Li et al., 2022), though primarily a detection framework, is extensible to segmentation tasks via mask prediction heads and global receptive fields from its transformer backbone. Mask2Former (Cheng et al., 2022a) unify semantic, instance, and panoptic segmentation using masked attention and multi-scale features. These models excel in scenarios where object boundaries are complex and category sets are large.

Our benchmark includes these three representative architectures to evaluate segmentation performance on fine-grained insect masks. The inclusion of expert-reviewed instance contours in LabInsect-48K facilitates rigorous evaluation of morphological segmentation, which is critical for extracting traits like mask-based aspect ratio, wing area, and convexity.

## 3 OUR LABINSECT-48K DATASET

### 3.1 DATA COLLECTION

LabInsect-48K is built from digitized insect specimens curated by the Australian National Insect Collection (ANIC), the world's largest collection of Australian insects and related groups such

Figure 2: Examples of annotations in our LabInsect-48K dataset. For each sample, we provide the biology taxonomy, the bounding box, and the mask.

as mites, spiders, earthworms, nematodes and centipedes. All specimens are photographed using high-resolution imaging systems under standardized conditions, including controlled illumination and fixed camera distance. Each image contains a single specimen, carefully mounted to ensure a canonical dorsal view and minimal occlusion. This setup preserves fine anatomical details, *e.g.*, wing venation, leg articulation, and antennae. The initially obtained dataset comprise more than 80,000 images, covering four major insect orders.

## 3.2 DATA CLEANING AND PREPROCESSING

We performed multi-stage quality control to remove corrupted, low-quality, or duplicate images. Two human annotators independently reviewed all images to ensure specimen completeness, verifying that key anatomical structures such as antennae, wings, and body contours were clearly visible. Annotators also compared image pairs to identify and remove near-duplicates. Meanwhile, images with ambiguous or missing taxonomic metadata are excluded. After filtering, each retained image contains a single specimen accompanied by complete and verified textual annotations, including clearly defined genus and species labels. To aid model generalization, images were retained in their original high resolution (mostly above $4000 \times 5000$ pixels).

## 3.3 DATA ANNOTATION

**Semantic Annotations.** To construct a taxonomically meaningful dataset, we assign each image a three-level hierarchical label: *order*, *genus*, and *species*. To organize the dataset, images are grouped by insect order into separate directories, facilitating efficient taxonomy-aware processing.

To extract semantic labels from the specimen metadata embedded in the images (*e.g.*, specimen tags), we adopt PaddleOCR (PaddleOCR, 2020). It is an ultra lightweight optical character recognition (OCR) system, supporting over 80 languages recognition. Compared to heavier LLM-based vision-language models, PaddleOCR operates efficiently with minimal GPU resources, thus providing a practical solution for large-scale, resource-constrained annotation workflows. It consistently delivers accurate recognition on standardized specimen labels under the collected images.

For each image, we parse the extracted text to identify critical semantic fields: the *genus_species* name, collection locality, collection date, collector's identity, ANIC catalog number, and size reference metrics. Among these, we retain only the *genus_species* categorization for downstream taxonomic tasks. Images where PaddleOCR fails to extract legible or valid taxonomic information are discarded to ensure the integrity and reliability of semantic labels. This semi-automated approach allows us to generate large-scale, expert-aligned annotations with high precision while minimizing manual effort.

**Bounding Boxes in Spatial Annotations.** We employ a hybrid semi-automated pipeline to generate high-quality bounding box annotations. First, we use Grounding DINO (Liu et al., 2023) with class-agnostic prompts (*e.g.*, *"insect"*) to predict initial bounding boxes for each specimen. These initial predictions are manually refined using X-AnyLabeling (Wang, 2023), a human-in-the-loop annotation tool that allows quick correction of inaccurate or missed detections. To accelerate large-scale labeling, we then train a YOLOv8 (Varghese & Sambath, 2024) model on this corrected subset and use it to re-predict bounding boxes on the remaining unlabeled images. YOLOv8 is chosen over Grounding DINO for this stage due to its superior inference speed and sufficient accuracy in this context. The corrected predictions are reviewed and refined, and the improved annotations are fed back to retrain

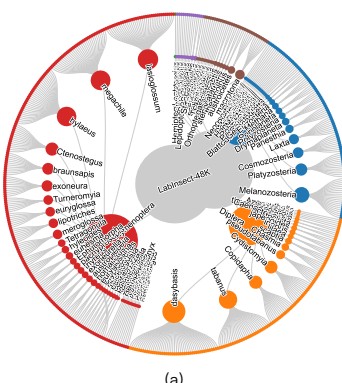 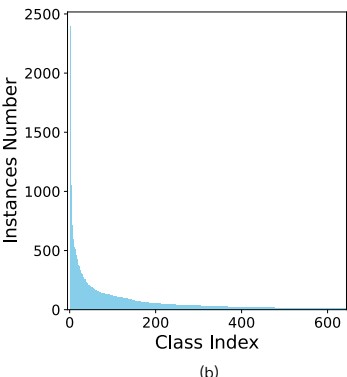

(a) (b)

Figure 3: Statistics of our LabInsect-48K dataset. (a) The hierarchical structure of our dataset; (b) The long-tailed distribution of our dataset.

the detector in an iterative loop. After multiple refinement rounds, we obtain accurate and consistent bounding box annotations across the entire dataset.

**Masks in Spatial Annotations.** Given the bounding boxes, we use them as spatial prompts to initialize segmentation mask prediction using Segment Anything Model (SAM) (Kirillov et al., 2023). SAM typically produces high-quality masks under clean imaging conditions, even when specimens are not centrally positioned. However, in cases where SAM outputs are suboptimal, potentially due to pose variation, small body parts, we further correct the masks using an interactive annotation mode. Annotators manually placed additional input points as corrective prompts to refine SAM's predictions. This human-in-the-loop refinement ensured precise contour fidelity, including critical structures such as antennae, wing tips, and legs. All finalized masks undergo a two-pass verification process and are stored in COCO-compatible RLE formats for downstream training.

This multi-stage annotation process balances automation with manual precision and allows LabInsect-48K to provide fine-grained, high-quality spatial annotations necessary for detection, segmentation, and morphological analysis (see some examples in Figure 2). After completing the annotation workflow, we further apply a filtering step to exclude species represented by fewer than 10 images, in order to ensure meaningful train/val/test splitting. The total size of the final dataset is approximately 240 GB on disk. It consists of 48,400 images spanning 4 orders, 123 genus, and 643 species. The overall data distribution exhibits a naturally long-tailed pattern, with a few dominant species and many rare ones (see Figure 3 (b)). Such a distribution reflects the ecological realities of insect biodiversity.

### 3.4 DATASET STRUCTURE

LabInsect-48K provides two formats to accommodate both per-image annotation format and large-scale training pipelines. **(1) Folder-Based Structure.** The primary format organizes images by taxonomic hierarchy. At the top level, directories correspond to insect orders. Within each order, subdirectories represent genus-species pairs (*e.g.*, `Lepidoptera/Helicoverpa_armigera/`). Each image is stored in JPG format and is accompanied by a dedicated JSON file with the same filename (see Figure 3 (a)). These per-image JSON files contain all annotation fields: semantic labels, bounding box coordinates, and instance masks encoded in RLE format. This structure allows for flexible inspection, visualization, and editing at the single-sample level. **(2) COCO-Style Flat Format.** To support large-batch training and compatibility with standard detection and segmentation frameworks, we also provide a consolidated version of the dataset. In this format, all images are stored in a single directory, and a unified COCO-style JSON file contains annotations for all samples. Each entry includes image metadata, category labels, bounding boxes, and instance masks. This structure enables seamless integration with popular toolkits such as MMDetection (Chen et al., 2019), Detectron2 (Wu et al., 2019b), and YOLO-based pipelines (Varghese & Sambath, 2024). Both formats contain consistent and aligned annotations. Users may choose the structure that best fits their training, evaluation, or visualization needs.

Considering the full dataset size ( 240 GB on disk), we additionally provide a curated subset, *i.e.*, LabInsect-48K-subset, for faster experimentation and benchmarking. This subset is created by randomly sampling approximately 20% of the images from each species in the full dataset, resulting

Table 2: Results of fine-grained classification on our dataset.

| Model | Top-1 Acc | Macro-Precision | Macro-Recall | Macro-F1 |
|---|---|---|---|---|
| ResNet50 (He et al., 2016) | 0.9831 | 0.9855 | 0.9770 | 0.9770 |
| HRNet-W32 (Sun et al., 2019) | 0.9836 | 0.9834 | 0.9795 | 0.9779 |
| ViT-B/16 (Dosovitskiy et al., 2021) | 0.9096 | 0.9039 | 0.8898 | 0.8822 |
| MobileNetV4-L (Qin et al., 2024) | 0.9947 | 0.9941 | 0.9916 | 0.9913 |
| CLIP Zero-Shot (Radford et al., 2021) | 0.1974 | 0.1741 | 0.1868 | 0.1457 |

Table 3: Results of object detection methods on our dataset.

| Method | Backbone | AP | AP$^{50}$ | AP$^{75}$ | AP$^{M}$ | AP$^{L}$ | AR | AR$^{M}$ | AR$^{L}$ |
|---|---|---|---|---|---|---|---|---|---|
| Faster RCNN (Ren et al., 2015) | HRNetV2 | 0.474 | 0.560 | 0.551 | 0.366 | 0.476 | 0.707 | 0.366 | 0.715 |
| DETR (Carion et al., 2020) | ResNet50 | 0.126 | 0.163 | 0.095 | 0.133 | 0.157 | 0.263 | 0.214 | 0.261 |
| YOLOX (Ge et al., 2021) | YOLOX-L | 0.161 | 0.185 | 0.180 | 0.237 | 0.160 | 0.309 | 0.236 | 0.310 |
| Deformable DETR (Zhu et al., 2020) | ResNet50 | 0.699 | 0.794 | 0.779 | 0.465 | 0.706 | 0.874 | 0.514 | 0.882 |
| DINO (Zhang et al., 2022) | ResNet50 | 0.712 | 0.757 | 0.749 | 0.537 | 0.717 | 0.932 | 0.618 | 0.940 |
| ViTDet (Li et al., 2022) | ViT-B | 0.437 | 0.600 | 0.553 | 0.338 | 0.443 | 0.643 | 0.374 | 0.650 |
| DiffusionDet (Chen et al., 2023) | ResNet50 | 0.438 | 0.478 | 0.471 | 0.469 | 0.442 | 0.906 | 0.570 | 0.913 |
| Mask R-CNN (He et al., 2017) | ConvNeXt-V2 | 0.541 | 0.678 | 0.649 | 0.358 | 0.547 | 0.757 | 0.404 | 0.766 |

in a reduced version of around 40 GB. Importantly, the subset preserves the same taxonomic categories (orders, genera, and species) as the full dataset, differing only in the number of images per category. For consistency, we also offer this subset in both the folder-based and COCO-style formats, allowing users to prototype and validate models before scaling up to the full dataset.

## 4 EXPERIMENTS

To evaluate the utility of LabInsect-48K in supporting various visual recognition tasks, we design a series of benchmark experiments across fine-grained classification, object detection, and instance segmentation. Given the large size of the full dataset, we conduct our experiments on the curated subset (40 GB), which preserves the full taxonomic label space while offering a more computationally efficient evaluation environment.

### 4.1 FINE-GRAINED CLASSIFICATION

Our dataset supports species-level classification and open-vocabulary analysis. In real-world biodiversity monitoring, many species encountered during deployment may not be seen during training. To explore this scenario, we evaluate both closed-set classifiers and a zero-shot open-vocabulary model.

**Task Definition.** We focus on species-level classification, where models predict the exact species name from a predefined label set. We also examine the ability of CLIP (Radford et al., 2021) to generalize to unseen categories using species-level textual prompts (*e.g.*, "a photo of insect species $X$"). We report standard metrics including top-1 accuracy, macro-precision (Manning et al., 2008), macro-recall (Manning et al., 2008), and macro-F1 (Manning et al., 2008).

**Models.** We benchmark a diverse set of architectures: ResNet-50 (He et al., 2016), HRNet-W32 (Sun et al., 2019), ViT-B/16 (Dosovitskiy et al., 2021), MobileNetV4-L (Qin et al., 2024), and CLIP in zero-shot mode. These models span both convolutional and transformer-based paradigms.

**Results.** Table 2 summarizes the performance of all models. Among closed-set classifiers, HRNet-W32 and MobileNetV4-L achieve the highest accuracy and macro-F1 scores, showing strong performance on this fine-grained task. From the entomological perspective, such strong baseline performance is in fact a positive feature: it enables reliable initial species-level labeling, making the dataset highly valuable for real-world applications such as automated specimen pre-sorting, rapid species triage, and large-scale digitization of entomological collections. CLIP, evaluated in zero-shot mode without access to training data, achieves limited accuracy. Its relatively low performance highlights the challenge of applying open-vocabulary models to fine-grained biological datasets where domain-specific features and visual cues are crucial.

Table 4: Instance segmentation results on our LabInsect-48K dataset.

| Method | Backbone | mIoU | mAcc | mFscore | maskAP |
|---|---|---|---|---|---|
| Mask RCNN (He et al., 2017) | Swin-T | 0.8071 | 0.8421 | 0.8481 | 0.514 |
| Mask RCNN (He et al., 2017) | ConvNeXt-V2 | 0.8089 | 0.8443 | 0.8475 | 0.579 |
| BoxInst (Tian et al., 2021) | ResNet50 | 0.8094 | 0.8474 | 0.8614 | 0.217 |
| Decoupled-SOLO (Wang et al., 2020b) | ResNet50 | 0.8428 | 0.8969 | 0.8890 | 0.238 |
| SOLOv2 (Wang et al., 2020c) | ResNet50 | 0.8844 | 0.9271 | 0.9160 | 0.476 |
| QueryInst (Fang et al., 2021) | ResNet50 | 0.8786 | 0.9190 | 0.9227 | 0.229 |
| ViTDet (Li et al., 2022) | ViT-B | 0.7840 | 0.8145 | 0.8259 | 0.503 |

## 4.2 INSECT LOCALIZATION AND DETECTION

We assess the capability of object detection models to localize and classify insect specimens from high-resolution images. The bounding box annotations in our dataset allow models to extract region-of-interest features under varying body sizes and insect morphology.

**Experimental Setup.** We benchmark a range of closed-set object detectors, including Faster R-CNN (Ren et al., 2015), YOLOX (Ge et al., 2021), DETR (Carion et al., 2020), Deformable DETR (Zhu et al., 2020), DINO (Zhang et al., 2022), ViTDet (Li et al., 2022), DiffusionDet (Chen et al., 2023), and Mask R-CNN (He et al., 2017). All models are trained on the curated subset with default backbones as shown in Table 3, and evaluated using COCO-style metrics: average precision (AP) and average recall (AR) across multiple IoU thresholds and object sizes.

**Results.** Table 3 reports the performance of all methods. DINO and Deformable DETR yield the best results, achieving AP scores of 0.712 and 0.699 respectively, demonstrating strong robustness to fine-grained insect variation. DiffusionDet and Mask R-CNN also perform well, particularly in recall, with AR values above 0.75, indicating high sensitivity to object presence. Faster R-CNN and YOLOX deliver moderate performance, especially in $AP^{50}$ and $AP^{75}$, indicating their ability to capture bounding boxes with relaxed thresholds. ViTDet shows moderate performance, suggesting transformer-based backbones benefit from higher resolution but require careful tuning. Overall, our results highlight that transformer-based detectors equipped with deformable attention, such as DINO and Deformable DETR, consistently outperform traditional convolutional and vanilla transformer models, making them more suitable for biodiversity-focused detection tasks.

## 4.3 INSTANCE SEGMENTATION TASK

To enable pixel-level morphological analysis, we evaluate a set of instance segmentation models on our dataset. Unlike detection, segmentation requires models to capture precise object boundaries, which is crucial for downstream applications such as trait quantification, shape-based clustering, and computational taxonomy.

**Experimental Setup.** We evaluate representative methods across classical, region-based, and query-based paradigms, including Mask R-CNN (He et al., 2017), BoxInst (Tian et al., 2021), SOLOv2 (Wang et al., 2020c), Decoupled-SOLO (Wang et al., 2020b), QueryInst (Fang et al., 2021), and ViTDet (Li et al., 2022). Backbones span Swin Transformer, ConvNeXt, and ResNet variants. Models are trained using a combination of binary cross-entropy and Dice loss, and evaluated using mean Intersection-over-Union (mIoU), mean accuracy (mAcc), mean F1-score (mFscore), and mask average precision (maskAP).

**Results.** Table 4 presents detailed performance metrics. SOLOv2 achieves the highest mIoU (0.8844), mAcc (0.9271), and mFscore (0.9160), demonstrating its strength in precise spatial delineation. QueryInst also performs strongly in overall F-score, likely due to its iterative refinement strategy. Mask R-CNN with ConvNeXt-V2 backbone shows balanced performance across all metrics and achieves the best maskAP (0.579), indicating its capacity to recover complex boundaries and fine structures. Interestingly, BoxInst and Decoupled-SOLO attain high mIoU and mAcc but exhibit lower maskAP, suggesting they are effective at spatial coverage but less accurate in contour sharpness or fine part segmentation. ViTDet underperforms across most metrics, implying limitations of its pretrained features when applied to detailed mask prediction in biological images.

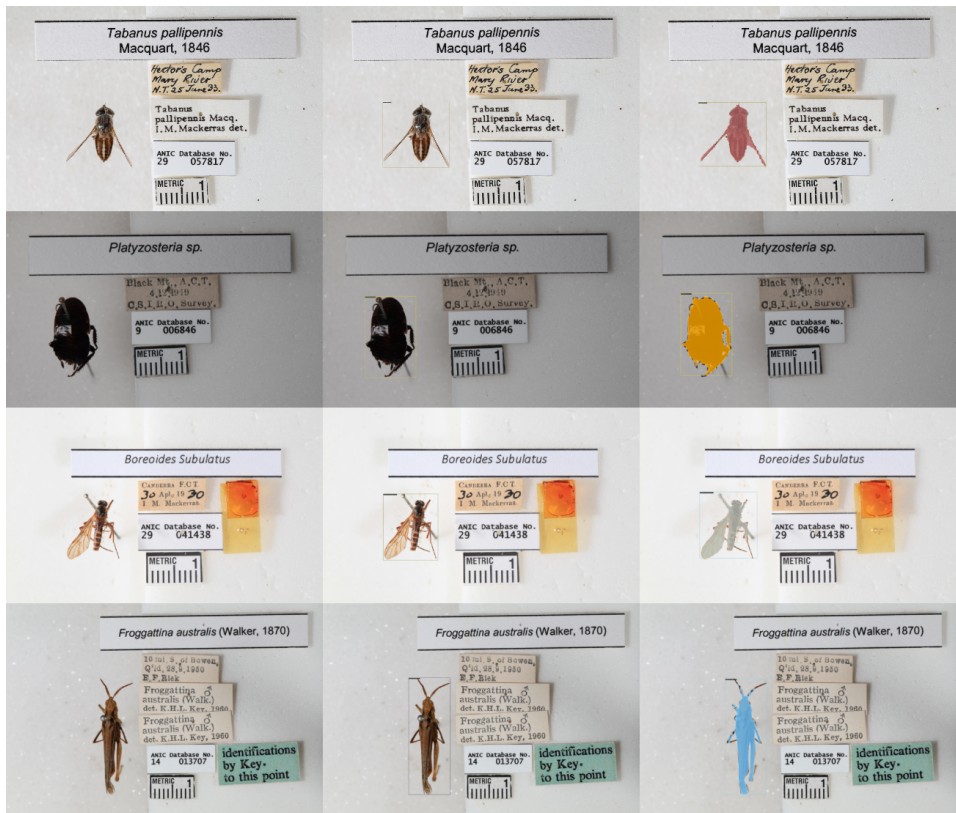

Figure 4: Qualitative detection and segmentation results. From left to right: original image, bounding box predicted by DINO, and instance mask predicted by Mask R-CNN.

Overall, these results emphasize that top-down and mask-driven architectures, particularly those with region-free or dynamic kernels, better capture specimen boundaries essential for biodiversity analysis. The inclusion of expert-labeled masks in our dataset enables rigorous benchmarking of these models and facilitates further research into morphology-aware learning.

### 4.4 QUALITATIVE EVALUATION

Figure 4 visualizes some bounding box predictions and mask predictions. From left to right, we show the original image, bounding box predicted by DINO, and instance mask predicted by Mask R-CNN. While the overall mask quality is high, we observe visible inaccuracies in fine structures such as antennae and legs. This highlights the challenge of capturing intricate morphology even with high-resolution inputs, and underscores the need for structure-aware segmentation refinement in future research.

## 5 CONCLUSION

In this paper, we introduce LabInsect-48K, a comprehensive, high-resolution dataset designed to support multi-task insect understanding through classification, detection, and segmentation. Compared with existing insect datasets, LabInsect-48K provides both semantic and spatial annotations, enabling not only species recognition but pave the way for trait-level analysis through detailed instance masks. We benchmark a variety of state-of-the-art vision models across all tasks and reveal both strengths and limitations of current approaches when applied to fine-grained biological data. The study on open-set discovery and cross-dataset generalization validate that our dataset facilitates accurate, fine-grained, and interpretable insect recognition and analysis for entomological research.

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
