Table 1: Training configurations of object detection models evaluated on LabInsect-48K-subset. 'LD': learning rate; 'WD': weight decay.

| Method | Backbone | Epochs / Iter. | Batch Size | Optimizer | Hyperparameters |
|---|---|---|---|---|---|
| Faster R-CNN | HRNetV2-W32 | 12 epochs | 2 | SGD | LR = 0.02, momentum = 0.9, WD = 1e-4 |
| DETR | ResNet-50 | 40 epochs | 2 | AdamW | LR = 1e-4, WD = 1e-4 |
| YOLOX | YOLOX-L | 20 epochs | 8 | SGD (Nesterov) | LR = 0.01, momentum = 0.9, WD = 5e-4 |
| Deformable DETR | ResNet-50 | 30 epochs | 16 | AdamW | LR = 2e-4, WD = 1e-4 |
| DINO-4Scale | ResNet-50 | 12 epochs | 2 | AdamW | LR = 1e-4, WD = 1e-4 |
| ViTDet | ViT-B | 40k iters | 4 | AdamW | LR = 1e-4, WD = 0.1 |
| DiffusionDet | ResNet-50 | 50k iters | 2 | AdamW | LR = 2.5e-5, WD = 1e-4 |
| Mask R-CNN | ConvNeXt-V2 | 36 epochs | 4 | AdamW | LR = 1e-4, WD = 0.05 |

Table 2: Training configurations of instance segmentation models on LabInsect-48K-subset. 'LD': learning rate.

| Method | Backbone | Epochs / Iter. | Optimizer | Hyperparameters |
|---|---|---|---|---|
| Mask R-CNN | Swin-T | 12 epochs | AdamW | LR = 1e-4, betas = (0.9, 0.999), weight decay = 0.05 |
| Mask R-CNN | ConvNeXt-V2 + FPN | 36 epochs | AdamW | LR = 1e-4, betas = (0.9, 0.999), weight decay = 0.05 |
| BoxInst | ResNet-50 + FPN | 50k iters | SGD | LR = 0.01, momentum = 0.9, weight decay = 1e-4 |
| Decoupled SOLO | ResNet-50 + FPN | 12 epochs | SGD | LR = 0.01, momentum = 0.9, weight decay = 1e-4 |
| SOLOv2 | ResNet-50 + FPN | 12 epochs | SGD | LR = 0.01, momentum = 0.9, weight decay = 1e-4 |
| QueryInst | ResNet-50 + FPN | 12 epochs | AdamW | LR = 1e-4, weight decay = 1e-4 |
| ViTDet | ViT-B (MAE pretrained) | 40k iters | AdamW | LR = 1e-4, betas = (0.9, 0.999), weight decay = 0.1 |

## A  USE OF LLMS

Large Language Models (LLMs) were used solely to refine the writing of the manuscript. We used an LLM strictly for linguistic improvements: refining wording, checking grammar, and smoothing the flow of the manuscript. The LLMs did not participate in the ideation, research methodology, or experimental design.

## B  IMPLEMENTATION DETAILS

**Fine-grained classification:** In the main paper, we primarily adopt ResNet50, HRNet-W32, ViT-B/16, MobileNetV4-L in fine-grained classification. Each model is trained with Adam optimizer for 100 epochs. The batch size is 16 and learning rate of $1 \times 10^{-3}$. All models were initialized with ImageNet-based pretrained weights, enabling more efficient convergence and leveraging prior knowledge from large-scale datasets. For CLIP Zero-Shot, we adopt the same configurations in openai/CLIP to do the inference.

**Object detection:** We adopt the MMDetection (Chen et al., 2019) framework to implement and evaluate some methods in our dataset. We select a diverse set of representative detectors, spanning both convolutional and transformer-based architectures, to ensure comprehensive evaluation. Most of our experiments are conducted using the default training configurations provided by MMDetection, with minor modifications made to certain hyperparameters for specific models. The detailed training settings for each model are summarized in Table 1.

**Instance segmentation:** To benchmark on the segmentation task, we also adopt the instance segmentation implementations in MMDetection (Chen et al., 2019). These models encompass a variety of design paradigms, including anchor-based, anchor-free, query-based, and transformer-based frameworks. Their inclusion enables a holistic assessment of segmentation performance under diverse architectural biases. The detailed training settings for each model are summarized in Table 2.

## C  DATASET SPLIT

To support fair evaluation across classification, detection, and segmentation tasks, LabInsect-48K is divided into non-overlapping training, validation, and test sets. Splits are constructed at the *species* level to avoid specimen-level leakage. We apply a fixed split ratio of 75%/10%/15% (train/val/test) within each species, ensuring that all three splits contain examples from each category. This per-class

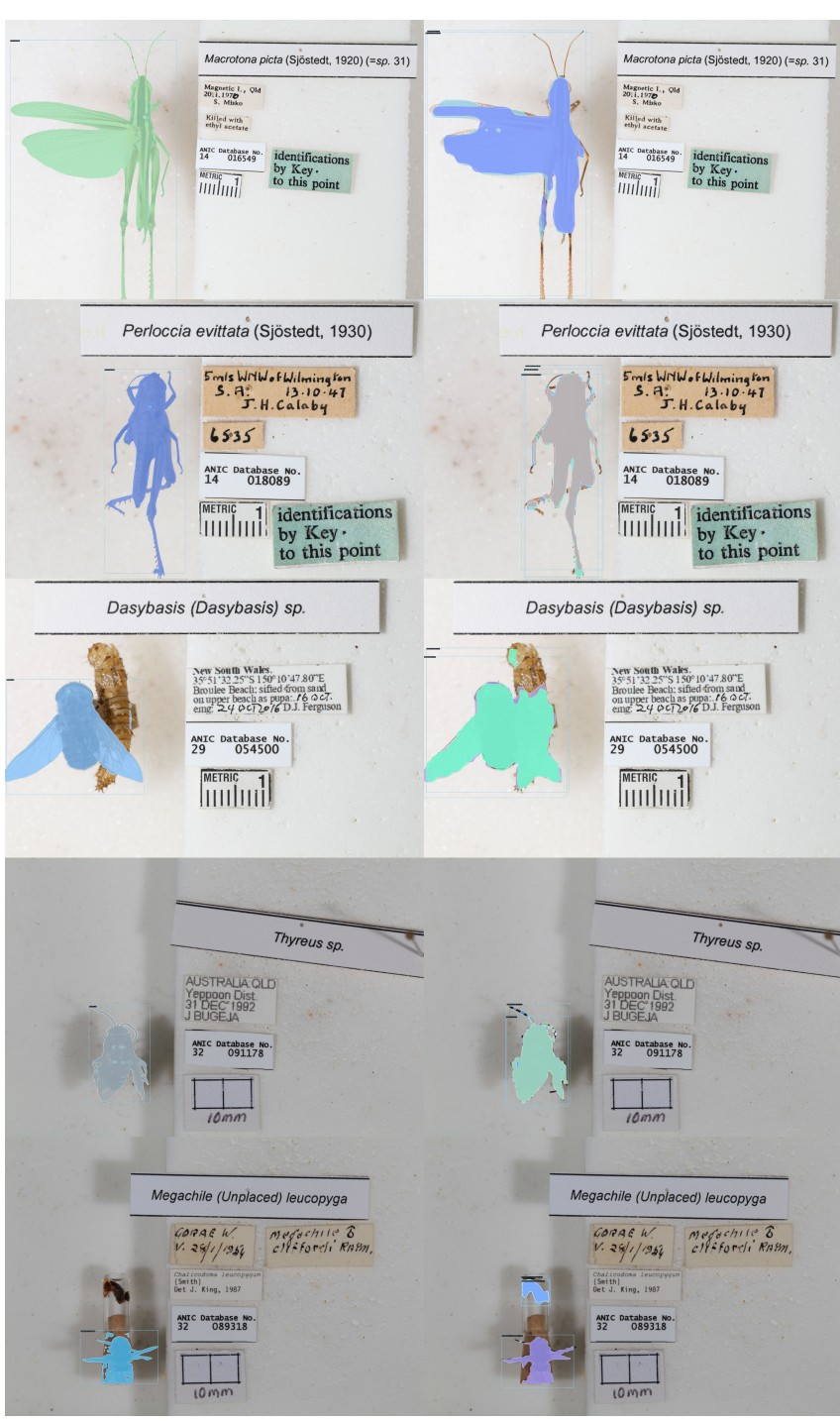

Figure 1: Some bad predictions from Mask R-CNN. Left: ground-truth; Right: predictions.

stratified sampling ensures that rare classes are preserved across partitions. The splits also maintain diversity in terms of insect shape, size and pose.

For the curated LabInsect-48K-subset, we adopt a simplified split strategy. Since the subset is intended for rapid benchmarking, we divide it into two parts only: 80% for training and 20% for testing. This split is also performed at the species level to preserve label integrity and maintain

Table 3: Top-1 Accuracy of different backbones on our dataset.

| Backbone | Top-1 Acc |
|---|---|
| ResNet50 (ImageNet pretrained) | 22.4% |
| HRNet-W32 (ours) | 42.6% |
| MobileNetV4-L (ours) | 48.2% |
| ResNet50 (ours) | 53.7% |

consistency with the full dataset. Separate annotation files are provided for each split to support reproducible experiments.

## D GENERALIZE TO UNSEEN CLASSES

To better assess the generalization ability of different models to novel, unseen insect species, we introduce a few-shot evaluation protocol on the IP102 dataset, which contains insect images captured in the wild with significant variation in lighting, background, and pose. Specifically, for each unseen class in IP102, we randomly sample 5 images as the support set, and use the remaining images as the query set. We extract visual embeddings using our supervised models (ResNet-50, HRNet-W32, and MobileNetV4-L, trained on our proposed dataset), compute the class prototypes by averaging the support embeddings per class, and then classify each query sample by retrieving the nearest class prototype in the embedding space using cosine similarity. We report the top-1 accuracy under this 5-shot setting in Table 3.

As seen above, the performance improves notably compared to ImageNet pretrained model. This new evaluation demonstrates the models' capabilities to generalize to newly discovered insect species, even with very limited labeled examples.

## E FAILURE CASES

We show several failure cases when applying existing models to our LabInsect-48K dataset here. We take Mask R-CNN (He et al., 2017) for example and visualize its bounding box and mask predictions in Figure. 1. These cases highlight limitations in both object localization and fine-grained segmentation in challenging insect images.

**Redundant and overlapping bounding boxes:** In multiple examples, Mask R-CNN produces repeated or overlapping bounding boxes for a single insect instance. This is likely due to insufficient non-maximum suppression (NMS) and high inter-class similarity across insect morphologies.

**Incomplete or coarse segmentation masks:** An obvious issue is the failure to segment fine structures such as: antennae, legs, and wing fringes. These elements are critical for biological interpretation and species-level discrimination. The coarse masks produced by Mask R-CNN are typically biased toward the body core and tend to miss thin or translucent parts. This can be attributed to: (1) The model's default RoI resolution being insufficient for fine details; (2) Loss of features due to aggressive downsampling in early CNN stages.

**Background mis-segmentation:** Some masks incorrectly include background regions, especially when the background contains insect eggs, fragmented body parts, or incomplete tissue from prior specimens. These visual distractions often confuse the model, causing it to involve parts of the background into the predicted mask. This compromises segmentation quality and can negatively affect downstream analyses, such as fine-grained morphological measurements.

**Class-specific nature of mask R-CNN masks:** Since Mask R-CNN applies a single mask head per class, rare or morphologically unique species may suffer from poor generalization. Unlike category-agnostic segmentation models like SAM, Mask R-CNN lacks a mechanism to refine segmentation based on visual cues and prompts, thus missing some details across categories.

These limitations suggest that traditional instance segmentation methods like Mask R-CNN may not be sufficient for high-fidelity insect understanding. Future improvements could involve: High-

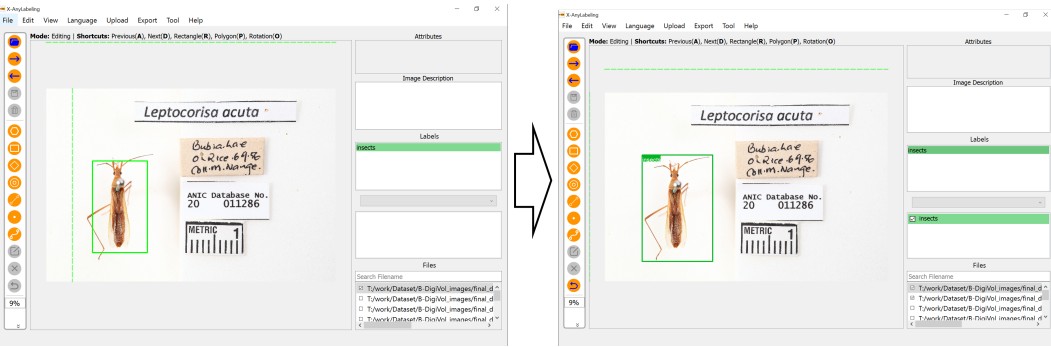

Figure 2: Human correction with X-AnyLabeling.

resolution mask heads or multi-scale refinement modules; Vision-language grounding, *e.g.*, GLIP, for better focus on biologically relevant regions.

## F  ANNOTATION PIPELINE AND CORRECTION WORKFLOW

**Bounding boxes**: Initial bounding boxes are generated using the open-vocabulary detector Grounding DINO. The annotations then go through two refinement stages:

- Stage 1: 10% of the samples are corrected by annotators using the X-AnyLabeling tool, which allowed precise editing, especially for small, occluded, or cluttered insects (see Figure 2).
- Stage 2: Model-assisted refinement, where a YOLOv8 detector trained on Stage 1 results is used to auto-suggest new boxes, which are further validated or corrected by annotators with X-AnyLabeling.

**Segmentation masks**: Similar to bounding boxes, we first generate masks with the Segment Anything Model (SAM). The bounding boxes obtained before are used as prompts. Then, we perform manual refinement to ensure accurate outlines of full-body insects and avoid background artifacts.

**Quality control**: (1) 10% of all annotations are independently double-checked by expert reviewers; (2) Overall, more than 700 annotation hours are used, involving 3 experienced annotators and 2 reviewers.

## G  DETAILS ABOUT ORDER AND GENUS

Table 4 summarizes the taxonomic structure of the LabInsect-48K dataset at the order and genus levels. Each insect order is listed alongside its corresponding genus, reflecting the broad taxonomic diversity covered by the dataset. Notably, the dataset spans major insect orders including Diptera, Blattodea, Orthoptera, Hymenoptera, Hemiptera, and Lepidoptera. The full genus-to-species mapping is not shown here since there are too many species. However, this detailed mapping is maintained within "genus_species_mapping.txt" file in the repository.

## H  COCO FORMAT SAMPLE ENTRIES

To ensure compatibility with widely used object detection and vision-language models, LabInsect-48K follows the standard COCO annotation format. This format organizes data into three key components:

**images:**

```
{
  "id": 0,
  "file_name": "20-011286.jpg",
```

Table 4: Insect orders and corresponding genus.

| Order | Genus |
|---|---|
| Hemiptera | Leptocorisa |
| Diptera | Anacanthella, Antissella, Austroplex, Boreoides, Caenoprosopon, Chasmia, Chrysomya, Copidapha, Cydistomyia, Dasybasis, Ectenopsis, Inopus, Japenoides, Lilaea, Pseudotabanus, Scaptia, Tabanus, Therevopangonia, Triclista |
| Blattodea | Anamesia, Ataxigamia, Blaberus, Calolampra, Celatoblatta, Cosmozosteria, Desmozosteria, Diploptera, Drymaplaneta, Eppertia, Euzosteria, Geoscapheus, Hemelytroblatta, Laxta, Litopeltis, Macrocerca, Megazosteria, Melanozosteria, Methana, Molytria, Nauphoeta, Neogeoscapheus, Neolaxta, Panesthia, Paranauphoeta, Periplaneta, Platyzosteria, Polyphagoides, Polyzosteria, Pycnoscelus, Scabina, Temnelytra |
| Orthoptera | Macrotona, Micreola, Sumbilvia, Aiolopus, Austracris, Austroicetes, Calephorops, Chortoicetes, Froggattina, Gastrimargus, Heteropternis, Maclystria, Macrazelota, Macrotona, Perloccia, Rusurplia, Schizobothrus, Stenocatantops, Sumbilvia, Tetrix, Theomolpus, Xypechtia |
| Hymenoptera | Austroplebeia, Austrothurgus, Brachyhesma, Braunsapis, Callohesma, Ceratina, Coelioxys, Euhesma, Euryglossa, Euryglossina, Euryglossula, Exoneura, Exoneurella, Homalictus, Hylaeus, Hyleoides, Hyphesma, Lasioglossum, Lipotriches, Megachile, Meroglossa, Nomia, Pachyprosopis, Palaeorhiza, Paracolletes, Sphecodes, Thyreus, Xanthesma, Agenioideus, Anoplius, Batozonellus, Ceropales, Cryptocheilus, Ctenostegus, Episyron, Ferreola, Hemipepsis, Heterodontonyx, Paracyphononyx, Pompilus, Telostegus, Tetragonula, Turneromyia, Heterodontonyx, Priocnemis, Turneromyia, Bothriomutilla, Ephutomorpha, Trogaspidia |
| Lepidoptera | Agrotis, Cosmoclostis, Exelastis, Hepalastis, Imbophorus, Megalorhipida, Pterophorinae, Sinpunctiptilia, Sphenarches, Stangeia, Stenoptilia, Stenoptilodes, Trichoptilus |

```
    "width": 5616,
    "height": 3744
}
```

**annotations:**

```
{
    "id": 0,
    "image_id": 0,
    "category_id": 0,
    "bbox": [x, y, width, height],
    "area": 467210.0,
    "segmentation": "",
    "iscrowd": 0
}
```

**categories:**

```
    {
        "id": 0,
        "name": "Leptocorisa_acuta",
        "supercategory": "Hemiptera"
    }
```

# I  LICENSE AND DATA USE

All images and annotations in LabInsect-48K are released under the Creative Commons Attribution-NonCommercial 4.0 International (CC BY-NC 4.0) license. This means the dataset is freely available for research and educational purposes. Commercial use is not permitted under this license.

The dataset sources include publicly accessible archives from the Australian National Insect Collection (ANIC). No private, sensitive, or restricted content is included. By sharing this dataset under a non-commercial license, we aim to support the academic communities while protecting against unauthorized commercial exploitation.

Users are encouraged to cite this work when using or adapting the dataset for non-commercial purposes, including model training, benchmarking, or scientific publication.