# OpenReview forum: "LabInsect-48K: A Comprehensive Dataset for Visual Insect Understanding"
_ICLR.cc/2026/Conference — ICLR 2026 Conference Withdrawn Submission_

### Official Review · Reviewer_Dnw6 · 2025-10-27

**Soundness:** 2
**Presentation:** 2
**Contribution:** 3
**Rating:** 4
**Confidence:** 4

**Summary:**

This paper introduces LabInsect-48K, a dataset consisting of 48,400 images across 643 insect species for fine-grained classification, object detection, and instance segmentation. It also benchmarks several relevant models by training and evaluating them on each of these tasks.

**Strengths:**

1. The paper introduces a new dataset for insects from museum collections, which has an important ecological application.
2. The dataset is not just for fine-grained classification, but includes instance segmentation masks, which is a great strength of the dataset.
3. The paper provides a clear description of the dataset structure and the smaller benchmark version improves accessibility for ML research.

**Weaknesses:**

My main concerns are about the dataset curation process and the fine-grained classification results. I will raise my score if the two major concerns are adequately addressed.

**Dataset Curation**
1. First, there is currently no way to review or inspect the dataset.
2. It is also unclear whether the semantic annotations were verified by humans. For the bounding box annotations, were all bounding boxes refined and validated by human annotators? How was the YOLOv8 model trained: on how many images, and was it trained on all species or only a subset? Similarly, for the bounding boxes and segmentation masks, were they reviewed by humans?

**Fine-Grained Classification Experiments**
1. It is unclear whether the reported accuracy values are percentages or raw proportions. Either way, the reported numbers are unusually high or low. I assume the values represent proportions (0 < acc < 1), in which case the reported 99% accuracy for MobileNet on such a fine-grained dataset seems extremely high. Please clarify whether these results correspond to training or test accuracy. If they are training accuracies, test accuracies should be reported instead. If they are indeed test accuracies, it is crucial to verify that there is no data leakage in the dataset split. Achieving such high accuracy with MobileNet on a fine-grained dataset seems unlikely.

    Also, does the model perform detection first, or does it make predictions directly on the whole image? If detection is not applied first, the model might rely on background information such as written labels to make a prediction. This will limit the dataset’s usefulness in identifying other unlabeled museum specimens, a well-suited use case of this dataset.

2. Since one of the main contributions of the paper is the hierarchical annotation, the results should also include genus-level and order-level classification performance. Additionally, it is unclear why family-level annotations are not provided. This should be clarified.

3. Besides, since CLIP is evaluated, it would be important to include BioCLIP [1] and CLIBD [2] as baselines as well. These models have the same architecture as CLIP and were trained with insect data, making them more appropriate and informative comparisons for this study.

    [1] Stevens, Samuel, et al. "Bioclip: A vision foundation model for the tree of life." Proceedings of the IEEE/CVF conference on computer vision and pattern recognition. 2024.

    [2] Gong, ZeMing, et al. "CLIBD: Bridging Vision and Genomics for Biodiversity Monitoring at Scale." The Thirteenth International Conference on Learning Representations.


**Minor issues**

1. The paper lists three main contributions: (1) dataset construction, (2) three-level annotations, and (3) benchmarking. However, the first two points together are one contribution (the introduction of the dataset). Regarding the third contribution, the paper only presents experimental results for fine-grained classification, detection, and segmentation. There are no experiments for open-set discovery or cross-dataset generalization, so these are not contributions of this work.

2. Related datasets such as Flatbug-dataset [1] and AMI [2] should be discussed and included in Table 1

3. For the detection and segmentation tasks, the paper does not evaluate Flatbug, a detection and segmentation model for arthropods, including insects [3]. How would a model specifically trained for insect detection and segmentation perform on this dataset?

    [1] Geissmann, Q.and A. Svenning. Flatbug-dataset a Compilation of Dataset of Terrestrial Arthropodes on Various Surfaces.       Zenodo, 29 Jan. 2025, https://doi.org/10.5281/zenodo.14761447.

    [2] Jain, Aditya, et al. "Insect identification in the wild: The ami dataset." European Conference on Computer Vision. Cham: Springer Nature Switzerland, 2024.

    [3] Svenning, Asger, et al. "A General Method for Detection and Segmentation of Terrestrial Arthropods in Images." bioRxiv (2025): 2025-04.


**Writing**
1. Figure 3(a): The figure is difficult to read. Please clarify what the four orders are. Some genus names are capitalized while others are not. Genus names should always be capitalized. Also, the color coding is unclear: is it based on the order level? There are more than four colors, which is inconsistent with the description in line 301.
2. Tables 2–4: Please specify whether the reported numbers represent percentage points or raw proportions.
3. Figure 1(c): The bottom-right image is upside down.
4. Figure 4: The bounding box color should be changed, as it is currently hard to see.
5. Line 256:  The “genus_species” name should be referred to as the scientific name (or binomial name).
6. The second listed contribution, “three-level annotations,” is somewhat confusing, as it seems to suggest the three taxonomic hierarchies: order, genus, and species. It would be clearer to say “three types of annotations.”

**Questions:**

1. How many insect orders are included in the dataset, and what are they? It is also unclear why there is no family-level annotation in this dataset.
2. How to derive “body length, wing area, mask-level aspect ratio, and compactness” directly from the masks (Line 86 - 87)? What are the definitions of mask-level aspect ratio and compactness?
3. It is unclear why this dataset can “facilitate interpretable insect recognition.” (Line 99)
4. What is the image size used for the experiments?

---

### Official Review · Reviewer_hAag · 2025-10-31

**Soundness:** 2
**Presentation:** 2
**Contribution:** 1
**Rating:** 0
**Confidence:** 5

**Summary:**

LabInsect-48K introduces a museum-style insect dataset comprising 48,400 high-resolution images across 643 species, providing hierarchical taxonomy (order/genus/species) along with spatial annotations (boxes and instance masks) to support classification, detection, and instance segmentation benchmarks. Data were curated under standardized imaging conditions; boxes were bootstrapped using Grounding DINO → YOLOv8, and masks were generated with SAM, followed by human refinement. Baselines span CNN/ViT classifiers, modern detectors (e.g., DINO, Deformable-DETR), and instance segmentation methods (e.g., Mask R-CNN, SOLOv2, QueryInst).

**Strengths:**

Useful spatial supervision for morphology: Compared with many insect datasets that are image-label only, the box + mask labels are valuable for trait extraction and fine-grained shape analyses.

Clean, standardized imaging: Canonical dorsal views at very high resolution make the set practical for controlled morphology tasks and method debugging.

Solid modern baselines: Inclusion of DINO / Deformable-DETR and strong mask models (e.g., Mask R-CNN, SOLOv2) provides reasonable reference points.

**Weaknesses:**

*Scale lags SOTA insect corpora*: At 48k images/643 species, LabInsect-48K is much smaller than contemporary benchmarks (e.g., BIOSCAN-5M: 5.15M images/specimens; Insect-1M: ~1.0M images; iNaturalist-2021: 2.7M train images over 10k species). This limits claims around “comprehensive” scale

*Comparable datasets already exist*. Large insect datasets (IP102, BIOSCAN-5M, Insect-1M) and some with detection boxes (IP102 provides ~19k bboxes) reduce the novelty of yet another classification/detection set; the mask addition is the main (perhaps minor) differentiator.

*Annotation pipeline is standard with current tools:* The use of Grounding DINO + SAM (widely adopted for rapid annotation) makes construction relatively straightforward; novelty lies more in curation than method.

*Venue fit (ICLR) is marginal:* The work is primarily a dataset paper with conventional benchmarks; conceptual ML novelty is limited for ICLR unless accompanied by new learning insights or tasks.

**Questions:**

Why not lean on self-/unsupervised labelers? Given FreeSOLO/CutLER can yield class-agnostic masks/boxes from unlabeled images, what is the measured quality/cost benefit of your Grounding-DINO→SAM loop over these approaches on your imagery? A small head-to-head would strengthen the annotation story.

LMM/VLM usage. Since annotation already uses vision-language tools (Grounding DINO) and SAM, have you explored open-vocabulary detection (e.g., prompt by genus/species) or multimodal LLMs for trait extraction on your masks? Even a small study could show added value beyond classic detection/segmentation.

---

### Official Review · Reviewer_hL8X · 2025-11-01

**Soundness:** 3
**Presentation:** 2
**Contribution:** 3
**Rating:** 6
**Confidence:** 4

**Summary:**

This paper introduces LabInsect-48K, a dataset of over 48k high-resolution insect images sourced from museum archives. LabInsect-48K spans 643 species across 4 insect orders, providing both Linnean taxonomic labels for classification and instance mask/bounding box annotations for vision-based analyses of insect morphology. The authors provide benchmarks for 3 tasks/challenges: fine-grained classification, object detection, and instance segmentation, using a range of well-known architectures (including zero-shot classification with CLIP). Authors claim this is the first dataset that combines comprehensive annotations for visual insect understanding.

**Strengths:**

- Contributes a high-quality, well-annotated, and comprehensive dataset for studying and classifying insects.
- Well-thought out annotation processes that has human-in-the-loop annotation to ensure high quality annotations with maximum efficiency.
- Extensive benchmark experiments using a range of competitive/well-known architectures. Good evaluation of CLIP for fine-grained zero-shot classification.
- Provision of dataset in COCO-style flat format facilitates future work with this dataset.

This could be a valuable dataset contribution, given some additional work re-framing the proposed machine learning challenges as tasks that would be valuable for ecologists and the inclusion of key experimental details.

**Weaknesses:**

- Literature review is not sufficiently comprehensive. Works which are missing include AMI, ALUS, flatbug, and ArTaxOr, which to some extent tackle the spatial (vision) and taxonomic-based challenges addressed by this dataset. I note there are other recent works which similarly provide segmentation masks for insect imagery, e.g. MassID45. However, as these are concurrent works, they do not impinge on the novelty of this paper.
- Limited discussion on the broader ecological benefits of this dataset (e.g., the analysis of *pre-existing* museum collections that have NOT been analyzed by human experts, or collections that humans do not have the time to analyze).
- Use of object detection is weakly motivated. Instance segmentation makes sense for morphological/trait based analyses, but what value does object detection have? We are not counting specimens in images with many insects; rather, we appear to have the same general image of one insect with label information to the right (Fig 2).
- Descriptions of annotation verification procedures are insufficient in places (L255, L285).
- While experts have technically provided annotations via the specimen tags, it would be good to know whether expert annotators (i.e., taxonomists) have verified the final semantic, bounding box, and mask annotations.
- Reproducibility details for the classification, instance segmentation, and object detection tasks are lacking. Dataset splits/learning rates/epochs/batch sizes/GPU hardware are absent.
- Potential for open-set discovery and cross-dataset discovery are mentioned, but no experiments are performed to explore this possibility.
- While the zero-shot experiments using CLIP are informative, it would be interesting to show zero-shot experiments using the more domain specific models such as BioCLIP or BioCLIP2 as well.
- Tables 2-4: Please mention in the captions whether each table shows results for models that are trained from scratch, fine-tuned, linearly probed, or deployed zero-shot. It is helpful to have enough high-level information with the table so one can interpret it without having to find the corresponding part of the main text (some of these appear not to be described in the main text either).
- While it is good to see that the authors compare several architectures for each challenge, some discussion on the computational requirements and feasibility of different models would strengthen this paper.
- Given the (painstaking) efforts put into gathering high-resolution specimen images and verifying specimen completeness, some rudimentary experiments on morphological analysis would be quite valuable. For example, panoptic segmentation to identify and quantify insect parts, counting number of legs/wings etc.

**Minor**
- L018 Abstract contains a fragment/incomplete sentence: "Delivers comprehensive..."
- L052 It would be helpful to indicate the number of pixels per mm here (or similar) as well
- L125,142 I believe BIOSCAN-1M and 5M do have bounding boxes available for download, though they are machine generated rather than human annotations and not manually verified. BIOSCAN-5M, page 4, says "The bounding box of the cropped region is provided as part of the dataset release".
- L178 Joint-authorship asterisk should not be in author name for Li* et al (2022).
- Generalizability to in-the-wild samples is limited. This dataset does not account for factors like stage of life, partial insects, etc., that may be encountered in ecological studies. It is fine to have a dataset which is limited to lab settings, but this limitation should be noted.

**References**
- [ALUS](https://doi.org/10.1111/2041-210X.13769): Schneider et al (2021). "Bulk arthropod abundance, biomass and diversity estimation using deep learning for computer vision". Methods in Ecology and Evolution.
- [ArTaxOr](10.1186/s44147-023-00284-8): Mazen (2023). "Arthropod Taxonomy Orders Object Detection in ArTaxOr dataset using YOLOX".  J. of Engineering and Applied Science, 70.
- [AMI](https://arxiv.org/abs/2406.12452): Jain et al (2024) "Insect Identification in the Wild: The AMI Dataset"
- [flatbug](https://www.biorxiv.org/content/10.1101/2025.04.08.647223v1): Svenning et al (2025) "A General Method for Detection and Segmentation of Terrestrial Arthropods in Images".
- [MassID45](https://arxiv.org/abs/2507.06972): Orsholm et al (2025) "A multi-modal dataset for insect biodiversity with imagery and DNA at the trap and individual level". arxiv:2507.06972
- [BioCLIP](https://arxiv.org/abs/2311.18803): Stevens et al (2024). "BioCLIP: A Vision Foundation Model for the Tree of Life". CVPR 2024. arxiv:2505.23883
- [BioCLIP2](https://arxiv.org/abs/2505.23883): Gu et al, (2025). "BioCLIP 2: Emergent Properties from Scaling Hierarchical Contrastive Learning". arxiv:2505.23883

**Questions:**

- If the images contain genus and species information via the specimen tags already, wouldn't it be possible to "cheat" on the classification challenge via vision-language models? That being said, if the object detection is used to crop the insect from the museum image, then classifiers are trained on the cropped insect, that would reasonable (this also aligns with previous work like BIOSCAN-1M/5M and AMI).
- L255 Can you provide an estimate for the accuracy of the PaddleOCR method for extracting the labels for this dataset? How many samples were manually inspected and what was the error rate?
- What is the ecological significance of the 4 insect orders and 643 species included in this dataset? The introduction motivates biodiversity monitoring in general, but it would be useful to know the importance of the species which the data concerns.
- L323 Why did you name the mini dataset "LabInsect-48K-subset" instead of "LabInsect-10K"...? Perhaps it would be helpful if the subset rarefied species at different rates to flatten the long-tailed distribution and preserve the numbers available for the infrequent species?

---

### Official Review · Reviewer_ZaHm · 2025-11-02

**Soundness:** 2
**Presentation:** 3
**Contribution:** 1
**Rating:** 2
**Confidence:** 5

**Summary:**

The paper introduces LabInsect-48K, the dataset of images of the Australia National Insect Collection (ANIC) specimens, together with bounding boxes for each specimen and insect segmentation. The dataset is benchmarked on the standard species classification  task against vanilla models.

**Strengths:**

-	The dataset is well described and organized.
-	The additional annotations are potentially time and resource-saving in some downstream tasks.

**Weaknesses:**

-	The dataset is already part of GBIF, which is part TreeOfLife-200M (https://huggingface.co/datasets/imageomics/TreeOfLife-200M), released with the Gu et al. “BioCLIP 2: Emergent Properties from Scaling Hierarchical Contrastive Learning”, https://doi.org/10.48550/arXiv.2505.23883 and the associated tools for data processing, deduplication, etc.
-	Adding bounding boxes and segmentation is not sufficient for meangiful downstream biological tasks. I was expecting localized traits, which would be useful and which are now increasingly becoming part of the morphological annotations, especially for herbaria collections (e.g., LeafMachine https://www.leafmachine.org)
-	Calling the evaluation task “fine grain” is misleading. Fine grain would be trait-level classification. Species classification is the standard task. In general, evaluation on a biological dataset that claims it contains information that aids biological tasks should be on biologically relevant tasks (and include domain-specific methods, e.g., BioTrove, BioCLIP, etc.)
For example, VLM4Bio: A Benchmark Dataset to Evaluate Pretrained Vision-Language Models for Trait Discovery from Biological Images
https://neurips.cc/virtual/2024/poster/97668, https://arxiv.org/abs/2408.16176
-	FishVista is a good example of ML-ready biological dataset with taxonomy, localization and segmentation, evaluated on biological tasks. Kazi Sajeed Mehrab, M. Maruf, Arka Daw, Abhilash Neog, Harish Babu Manogaran, Mridul Khurana, Zhenyang Feng, Bahadir Altintas, Yasin Bakis, Elizabeth G Campolongo, Matthew J Thompson, Xiaojun Wang, Hilmar Lapp, Tanya Berger-Wolf, Paula Mabee, Henry Bart, Wei-Lun Chao, Wasila M Dahdul, Anuj Karpatne; Proceedings of the IEEE/CVF Conference on Computer Vision and Pattern Recognition (CVPR), 2025, pp. 24275-24285. https://openaccess.thecvf.com/content/CVPR2025/html/Mehrab_Fish-Vista_A_Multi-Purpose_Dataset_for_Understanding__Identification_of_Traits_CVPR_2025_paper.html
It is based on Fish-AIR, an AI-ready multipurpose dataset of fish specimen collectionс: Bakış, Y., Wang, X., Altıntaş, B., Jebbia, D., & Bart Jr, H. L. (2023). On image quality metadata, fair in ML, AI-readiness and reproducibility: Fish-air example. Biodiversity Information Science and Standards, 7, e112178.

**Questions:**

How would you motivate the utility of the dataset? Are domain scientists (biologists) involved in the project? If this is a collaboration with domain scientists, what are the downstream tasks they are interested in that would be based on this dataset?

---

### Note · Authors · 2025-11-19

I have read and agree with the venue's withdrawal policy on behalf of myself and my co-authors.